# An Italian Single-Center Genomic Surveillance Study: Two-Year Analysis of SARS-CoV-2 Spike Protein Mutations

**DOI:** 10.3390/ijms26157558

**Published:** 2025-08-05

**Authors:** Riccardo Cecchetto, Emil Tonon, Asia Palmisano, Anna Lagni, Erica Diani, Virginia Lotti, Marco Mantoan, Livio Montesarchio, Francesca Palladini, Giona Turri, Davide Gibellini

**Affiliations:** 1Department of Diagnostic and Public Health, Division of Microbiology, University of Verona, 37134 Verona, Italy; riccardo.cecchetto@univr.it (R.C.); emiltonon@gmail.com (E.T.); asia.palmisano@studenti.univr.it (A.P.); anna.lagni@univr.it (A.L.); erica.diani@univr.it (E.D.); davide.gibellini@univr.it (D.G.); 2UOC Microbiology, AOUI Verona, 37134 Verona, Italy; giona.turri@aovr.veneto.it; 3Department of Diagnostic and Public Health, Division of Hygiene and Preventive, Environmental and Occupational Medicine, University of Verona, 37134 Verona, Italy; marco.mantoan@univr.it; 4Hospital Management, University Hospital of Verona, 37126 Verona, Italy; livio.montesarchio@aovr.veneto.it (L.M.); francesca.palladini@aovr.veneto.it (F.P.)

**Keywords:** SARS-CoV-2, whole genome sequencing, spike protein, mutational analysis, genomic surveillance

## Abstract

The repeated occurrence of SARS-CoV-2 variants, largely driven by virus–host interactions, was and will remain a public health concern. Spike protein mutations shaped viral infectivity, transmissibility, and immune escape. From February 2022 to April 2024, a local genomic surveillance program in Verona, Italy, was conducted on 1333 SARS-CoV-2-positive nasopharyngeal swabs via next generation full-length genome sequencing. Spike protein mutations were classified based on their prevalence over time. Mutations were grouped into five categories: fixed, emerging, fading, transient, and divergent. Notably, some divergent mutations displayed a “Lazarus effect,” disappearing and later reappearing in new lineages, indicating potential adaptive advantages in specific genomic contexts. This two-year surveillance study highlights the dynamic nature of spike protein mutations and their role in SARS-CoV-2 evolution. The findings underscore the need for ongoing mutation-focused genomic monitoring to detect early signals of variant emergence, especially among mutations previously considered disadvantageous. Such efforts are critical for driving public health responses and guiding future vaccine and therapeutic strategies.

## 1. Introduction

Severe acute respiratory syndrome coronavirus 2 (SARS-CoV-2, *Betacoronavirus pandemicum*), responsible for the global COVID-19 pandemic, infected more than 777 million people and caused nearly 7 million deaths, according to the data of the World Health Organization (WHO) from January 2025.

SARS-CoV-2 has a genome composed of about 30,000 nucleotides, encoding for spike (S), envelope (E), membrane (M), and nucleocapsid (N) structural proteins; 16 non-structural proteins (NSP); and 7 confirmed accessory proteins [1,2,3].

Spike protein forms a trimeric glycoprotein on the viral envelope and, after its binding with the cellular receptor, is cleaved by furine-like proteases into S1 and S2 subunits. The S1 subunit contains an N-terminal domain (NTD) and a receptor-binding domain (RBD) that is responsible for the virus binding to the angiotensin-converting enzyme 2 (ACE2) receptor on the target host cell. The S2 subunit carries out the fusion of the viral envelope with the host cell membrane [2,4].

The SARS-CoV-2 genome, despite the exonuclease proofreading activity of the viral non-structural protein 14 (NSP14), has exhibited considerable genetic variability, particularly in its spike protein [5], due to nucleotide substitutions, insertions, and deletions. Nucleotide substitutions have been extensively studied, with special emphasis on structural changes in viral proteins, particularly on S protein, which could alter the virus’s overall fitness [6]. SARS-CoV-2 strains are classified into Phylogenetic Assignment of Named Global Outbreak (PANGO) lineages according to shared substitutions, phylogenesis, and possible recombination events [7].

The WHO has identified several SARS-CoV-2 variants of concern (VOCs) and variants of interest (VOIs) based on transmissibility, pathogenicity, and impact on vaccines. However, focusing only on lineage frequency does not take into account the additional statistical power that can be gained by analyzing the independent behavior of the single mutation in different lineages [8]. Moreover, mutation-based analysis could be advantageous for understanding the determinants of a lineage phenotype and, thus, the biology of transmission and pathogenesis, suggesting possible phenotype prediction for new lineages.

New viral variants are often characterized by a high rate of viral replication and typically exhibit mutations in their spike protein, which have the potential to enhance viral transmission, pathogenicity, and immune recognition [2].

The functional and epidemiological significance of specific genetic changes has been clearly established, such as the D614G spike substitution, which is correlated with increased SARS-CoV-2 viral loads [9,10]. Additional substitutions on S protein, such as N439R, N501Y, and E484K, have been linked to immune escape and to increased transmissibility due to higher affinity of the S protein for ACE2 binding [9,10,11]. Despite these cases, identifying relevant substitutions remains a complex challenge. The identification of specific genetic changes that contribute to the virulence and transmissibility of the virus requires continuous monitoring and accurate analysis of its evolutionary phenomenon. In this study, we performed a comprehensive mutational analysis of SARS-CoV-2 spike protein in samples collected over two years of routine genomic surveillance (from February 2022 to April 2024). The aim of this study was to determine and describe the prevalence and impact of mutations in the S protein.

## 2. Results and Discussion

The S glycoprotein is the most well-studied structural protein of SARS-CoV-2. It is a type I homotrimer membrane protein anchored to the viral membrane that plays an essential role in human host cell surface receptor recognition by binding to receptor ACE2 [12,13]. The multifunctional nature of the S protein, which includes target recognition, cellular entry, and endosome escape, makes it a significant target for vaccine and therapeutic development [14,15].

The SARS-CoV-2 Wuhan-Hu-1 strain S protein contains 1273 amino acid residues, divided into two different subunits: a receptor-binding fragment S1 (14–686), responsible for target recognition, and a membrane proximal S2 subunit (687–1273), involved in membrane fusion and endosomal escape [16,17]. The S1 subunit is inherently dynamic and can lead to its division into an NTD, an RBD, and two different C-terminal domains (CTD1 and CTD2). The S2 subunit is conformationally stable prior to receptor engagement and includes a fusion peptide (FP), a fusion-peptide proximal region (FPPR), a heptad repeat 1 (HR1), a central helix (CH), a connector domain (CD), a heptad repeat 2 (HR2), a transmembrane segment (TM), and a cytoplasmic tail (CT) [16,17,18]. Concerning S mutations, the Omicron variant emerged with 23 unique variations, and the number has increased continuously since then [19]. All sequences of this study belonged to the Omicron variant, and mutations identified along S protein are displayed in Figure 1 and Figure 2.

From February 2022 to April 2024, a total of 1333 SARS-CoV-2 sequences collected from the University Hospital of Verona were analyzed. We identified 314 aminoacidic substitutions, 88 with total prevalence above 1%. This investigation focused only on this second group.

Based on their dynamics, the spike mutations were classified into five distinct categories: fixed mutations (*n* = 17), consistently detected with stable prevalence throughout the study period (whose normalized monthly prevalence was equal to or greater than 90% throughout the entire observation period, allowing for no more than two consecutive months below 90%); emerging mutations (*n* = 27), initially absent but subsequently appearing and persisting over time (whose relative monthly prevalence was 0% for at least the first month of the study, followed by a sustained increase in relative monthly prevalence, eventually exceeding 90%) with an ascending trend; fading mutations (*n* = 4), present at the beginning of the surveillance but later disappearing (with a relative monthly prevalence above 1% for at least four consecutive months at the beginning of the surveillance, followed by a decline to 0%) with a descending trend; transient mutations (*n* = 32), characterized by a temporary rise in prevalence (above 1% for at least two consecutive months) followed by a sustained decline; and divergent mutations (*n* = 8), exhibiting fluctuations with multiple phases of appearance and disappearance that do not meet the criteria of any of the previous categories. A heatmap with the relative monthly prevalence of all spike mutations analyzed in this study is shown in Appendix A.

Correlation between substitutions and, specifically, within each group, is visualized with a Spearman correlation heatmap in Figure 3. The fixed and emerging groups show higher correlation values when compared to the other three.

The mutation trends described were manually verified using CoV-Spectrum. Additionally, Spearman’s correlation analyses were performed to compare the global trend of each mutation with that observed in this study. The results are shown in Figure 4 (panels A, B, C, and D). Almost all substitutions show a positive correlation index. Mutations with values below 0.1 were further investigated and despite their low correlation, similar tendencies between local and global monthly prevalences can be observed. Appendix A compares their local and global trends.

### 2.1. Fixed Mutations

We classified fixed mutations as those observed during the period of analysis that did not change significantly in frequency in our population, so they are considered to have been stably acquired. A total of 17 mutations fall under this group. Among those permanent substitutions, only two involve the NTD (A27S, G142D), while eight seem to have permanently modified the RBD (S371F, S373P, S375F, T376A, D405N, R408S, K417N, N440K), three are in the CTD2 (D614G, H655Y, N679K), two are in the S2 cleavage site (N764K, D796Y), and the remaining two are located in the HR1 region (Q954H, N969K), as reported in Table 1.

G142D was originally associated with the Delta variant and faded transiently to rapidly reappear with Omicron onset. This substitution could be related to increased viral load and immune escape [20]; the latter is probably responsible for the introduction of A27S substitution [21], with a consistent presence in Omicron variants.

Substitution S373P, together with S375F, T376A, D405N, and R408S, appeared in November 2021 in a cluster of new mutations associated with the spread of Omicron BA.1. S373P is known to stabilize the β sheet core of RBD and to promote binding to the ACE2 receptor [22]. Interestingly, S373P and S375F appeared together with S371L, which was replaced by the stably acquired S371F mutation after a few weeks. S371F has been shown to interact with human palmitoyl transferase ZDHHC5 and accessory protein GOLGA7, both involved in cell entry [23]. Moreover, S371F, together with D405N and R408S, interferes with neutralizing antibodies [24]. R408S is also reported to be related to ACE2 binding and membrane fusion [25]. This mutation was not initially categorized as fixed, but further analyses using less stringent parameters (minimum depth, 30) revealed the presence of this mutation at above 90% prevalence for the whole period, matching the criteria for fixed mutations.

S375F and T376A significantly reduce S protein S1/S2 cleavage by TMPRSS2, and likely the pathogenicity of these strains as well [26].

The replacement of a lysine residue with an asparagine residue at position 417 (K417N) has been linked to reduced antibody neutralization and could reduce the binding strength due to the loss of a salt bridge, although this effect is compensated by other substitutions like N501Y or E484K [27]. Interestingly, neither of these two was found permanently in our population. RBD bearing N440K has lower binding affinity but better immune escape [28].

D614G, introduced in February 2020, was the first substitution to appear chronologically and is the only one that did not emerge concomitantly with the Omicron or Delta variants. This substitution confers increased binding to ACE2 receptor and viral replication [29].

The overall low fusogenicity and increased endosomal entry of the Omicron variant could be due to the H655Y substitution [30]. The N679K mutation is located in close proximity to the core region furin cleavage site (residues 682–685) and this aminoacidic replacement has been linked to increased furin-mediated cleavage [31]. N764K likely decreases the stability of S protein [32] and could also generate a potential cleavage site for SKI-1/S1P serine protease [33].

As for S2 substitutions, D796Y arose independently 35 times and appeared even before Omicron [34]. This mutation could offer a structural advantage by enhancing binding to TMPRSS2 [35], perhaps by altering the presentation of an immunogenic glycan epitope [36] and contributing to immune evasion [34]. Conversely, Q954H is critical in preventing antibody binding [37], while N969K is likely to be involved in reducing fusogenicity [38]. Table 1 summarizes all fixed mutations identified in this study, and their spatial disposition is shown in Figure 5.

### 2.2. Emerging Mutations

We defined an emerging mutation as one that was initially absent at the beginning of our analysis, appeared later during the investigated period, and persisted at high monthly prevalence until the study’s end [39,40,41,42]. All emerging mutations and their trends are presented in Figure 6.

The first group of substitutions, R21T, S50L, V127F, L216F, H245N, and A264D, are localized on the NTD of the S protein, and appeared around November 2023, drastically rising in prevalence. They are all associated with the BA.2.86 and JN.1 sublineages and might lead to altered antigenicity [39,43]. In detail, R21T causes a change in polarity in the residue (from positive to polar), but no structural modifications are reported. Specifically, S50L changes the residue polarity from polar to hydrophobic, and the S protein of BA.2.86 characterized by S50L is responsible for more efficient lung cell entry compared to BA.2 and EG.5.1 lineages, giving the virus a fitness advantage [44]. Mutation R158G was found in our study in low frequency in May 2023, rapidly rising then in December 2024. This mutation was already present in the Delta variant (June 2021–January 2022), linked to another fixed mutation G142D, correlated to weaker interactions with antibodies and increased viral load [20,45]. Mutation L216F does not change the residue polarity, which remains hydrophobic, and does not affect the infectivity in other lineages alone (B.1.1.7) [46]. However, L216F may be associated with other emerging mutations in new variants, which might lead to an increase in fitness. Mutations I332V, G339H, V445H, N450D, L452W, N460K, N481K, E484K, and F486P are located in the RBD of lineages of XBB.1.5 and BA.2.86 (I332V, V445H, N450D, N481K, and E484K only in BA.2.86). Due to their widespread prevalence, these mutations have been extensively evaluated to assess their impact. The I332V, G339H, V445H, N450D, N481K, and A484K mutations have been frequently investigated for their potential to increase immune evasion [47,48,49]. Moreover, L452W is involved in immune evasion from various mAbs and different vaccines [50].

L455S and F456L mutations are both located in the RBD region and seem to have a drastic impact on new lineages. The L455S acquired by JN.1 may change the binding affinity between RBD and the ACE2 receptor: studies demonstrate a notable reduction in ACE2 binding affinity for JN.1 RDB, while achieving the ability to evade RBD class 1, 2, and 3 antibodies. The F456L mutation, together with the transient L455F mutation, is one of the so-called “FLips”, named after the initials of the two mutated amino acid residues [51]. These mutations have appeared since May 2023 in different lineages (JG.3, EG.5.1.1, HV.1, and many more) and are associated with an increase in ACE2 affinity with a synergic combination, but a relationship between specific sublineages and increased clinical severity could only be assessed with epidemiological monitoring over time [52,53]. The strict parameters of the bioinformatic analysis explain the fall in the relative prevalence of the F456L mutation in January and February 2024. Further analyses with less stringent conditions revealed the presence of this mutation in these months as well.

S477N and T478K emerged in March 2022, are also associated with increased ACE2 binding affinity and are both located in the RBD. Specifically, according to molecular predictions, S477N decreases fluctuations stabilizing the S protein and can be considered the initial cause of the increased escape from the immune system through MHC-II [54]. T478K also enhances the stabilization of the RBD–ACE2 complex, but further investigations are needed [55,56]. Mutations Q498R, N501Y, and Y505H arose in March 2022, always associated with one of the others.

In Omicron’s VOI, there is a wide distribution of Q498R and Y505H mutations, while S447N and N501Y mutations are frequently present in Omicron subvariants with increased transmissibility [57,58,59]. On their own, Q498R and Y505H impact S protein in two different ways: negatively affecting both protein stability and bindings and significantly reducing its bonding to E37 of the ACE2. However, Q489R with N501Y and E484K increases ACE2 binding by around 50-fold compared with a WT protein [60], while the new Y505H forms a new favorable interaction on ACE2 K353 [61]. Studies on murine models also report how mutations Q498R and Y505H have unique adaptations to mouse ACE2, while N501Y adapted to both mouse and human ACE2 [58].

Mutation E554K is located on the CTD1 of the first subunit of S, which is a highly conserved domain. For this reason, many monoclonal antibodies (mAbs) are directed to this region and show broad and potent neutralization of many SARS-CoV-2 variants. This mutation in particular, first seen in B.1 in April 2020, maintained a low frequency throughout the pandemic, increasing in frequency in October 2023 in BA.2.86 and its sublineages, especially JN.1. E554K destabilizing the salt bridge formed with the triad K535, E554 and E583, disrupts both the interaction with the triad and all mAbs directed to this region, and confers improved antibody evasion to lineages carrying the mutation [62].

Instead, the other mutation on CTD1, A570V, emerged in November 2023 in our study with greater frequency in JN sublineages. The A570V mutation has been identified to have a lower spread, likely due to its lack of evolutionary advantage [63]. This mutation could have no direct effect on neutralization on mAbs targeting this domain [62], but instead it enhances the hydrophobic interactions between protomers, potentially increasing trimer stability [41]. However, more studies need to be conducted to assess the exact mechanisms on the stability of BA.2.86 S protein.

The P621S is a putative fusion-related mutation, already seen in some SARS-CoV-1 variants. This substitution occurs in the terminal region of the S1 and facilitates the formation of an α-helix in the 630 loop (residues 620–640), which is a key modulator for fusion, putatively impeding structural rearrangements for subsequent fusion [64,65].

The P681R mutation is the only emerging mutation found in the S1/S2 cleavage site, appearing in our study in October 2023. This mutation already appeared during the pandemic, in the first half of 2021 as a representative mutation of the Delta variant (lineage B.1.617). Previous studies demonstrated how P681R substitution could enhance the transmissibility of SARS-CoV-2 [59], as well as the cleavage of S protein and viral fusogenicity, explaining the higher pathogenicity of the Delta variant compared to the Wuhan lineage [66]. This mutation, which disappeared in January 2022 along with Delta, was also acquired by Omicron BA.2.86, conferring again the advantages of its ancestor to this lineage. S939F arose in November 2023, associated with new lineages JN and KP.3. This substitution slightly affects immune response but significantly increases the modulation of T-cells and the selective enrichment of potential binding epitopes for some HLA alleles [67].

The last emerging mutation, P1143L, which was found in our study from November 2023, is linked to Omicron XBB.1.5 and BA.2 subvariants. Even though this mutation has not been extensively studied, it has been reported in vitro that it could accelerate cell entry in pseudovirus, but in the context of actual human transmission, it could negatively affect spike stability [68,69]. As the P1143L mutation has been carried at high frequency since the beginning of 2024, more in vivo studies are needed to clarify whether lineages carrying this mutation have impaired human transmission or whether combinations with other mutations confer new fitness advantages [70].

All emerging mutations identified in this study are listed in Table 2, and their spatial disposition is shown in Figure 7.

### 2.3. Fading Mutations

Fading mutations are substitutions found at the beginning of the screening that disappeared during our study. Their trend is represented in Figure 8.

T19I mutation was found in our cohort consistently from February 2022 to February 2024, disappearing after March 2024. Worldwide, 6,102,968 sequences reported this mutation with an 84.64% overall prevalence. From January 2022 to May 2022, this mutation was reported in 15.8% of Omicron BA.2 sequences (21L clade) [71]. The BA.2-specific T19I mutation in NTD severely reduced S-mediated infection and processing, although it is not located near the S1/S2 furin cleavage site and does not affect known functional domains [72].

G339 is a major amino acid in RBD responsible for the interaction with the ACE2 receptor, and the G339D mutation is known to be involved in immune escape [73], and was reported to affect the neutralization performance of a subset of neutralizing antibodies [74]. This mutation disappeared in July 2023. In total, 5,429,192 sequences carrying this mutation were identified worldwide, with a 74.49% overall prevalence, mainly in sublineage BA.2 (21.60%). Consistent with our data, the mutation was reported to disappear in July 2023. The low prevalence between February and April 2022 is a consequence of strict sequencing parameters. Later analyses with a lower minimum depth (30) confirmed the presence of this mutation at high prevalence in these months.

Fading mutation Q493R, located in the RBD site, was last detected in our study in July 2022. As seen in the heatmap, this mutation had a low prevalence the first months of our study. However, this is a consequence of strict sequencing parameters, and further bioinformatic analyses with a lower minimum depth (30) confirmed the presence of this mutation for this period as well. A recent study [75] demonstrated a pivotal role of this mutation in the formation of new highly stable interactions (H-bond or salt bridge) with D38, E37, and E35 residues in ACE2 and some S1-RBD intrachain salt bridges, providing insight into the mechanism of increased binding of the Omicron spike S1 subunit to ACE2 and increased variant transmission rate [76]. Additionally, it was noted that the appearance of Q493R mutation can cause virological failure with bamlanivimab/etesevimab treatment. Non-responders to this treatment, particularly those who are immunocompromised, are advised to be checked for this mutation [77].

The P681H mutation in the SARS-CoV-2 glycoprotein in the polybasic furin cleavage site was acquired by the Alpha variant and it is supposed to improve site accessibility to furin, leading to enhanced cleavage and more efficient cell-to-cell fusion and syncytium formation [78,79]. It was reported that in human lung epithelial cells, the S protein of the Alpha variant confers a level of resistance to interferon-induced transmembrane protein (IFITM). In particular, the P681H mutation alone is sufficient to improve Alpha but not Omicron cell entry [79].

All fading mutations are listed in Table 3, and their spatial disposition is shown in Figure 9.

### 2.4. Transient Mutations

We define a transient mutation as one that appeared and then disappeared during our study period, with a single distinct peak in prevalence, particularly when there is a monthly prevalence above 1% for at least two consecutive months and then falls below 1% for more than two consecutive months. Transient mutations are shown in Figure 10.

The first transient mutation identified in the S protein is Q52H (first appearance April 2023, last appearance February 2024), located in NTD. This mutation, characteristic of EG.5.1, was also found with high prevalence in JG.3 and HV.1 and does not seem to provide any advantage to the virus, other than a likely increased resistance to cold inactivation [80].

The V83A and R346T mutations were present from January 2023 to February 2024 and from July 2022 to February 2024, respectively. Both these mutations are associated with increased fusogenicity, respectively, on XBB.1.5 and XBB.1 variants [81].

Mutation E180V was observed in this study from April 2023 to November 2023 and was mainly found in XBB.1.16, contributing to enhanced stability of the S protein [82].

H146Q mutation, which appeared in November 2022 and disappeared in February 2024, was mainly found in XBB.1.5. This residue is considered highly conserved and although the effects of the H146Q mutation are unclear, some studies reported that it is not involved in ACE2 binding or immune escape [83]. Moreover, this substitution does not change residue polarity and might not confer advantages in terms of fitness, likely a reason for the loss of this mutation in lineages after February 2024.

A cluster of NTD mutations, such as K147E, W152R, F157L, I210V, and G257S, were found with low frequency from November 2022 to January 2024. These mutations have been identified in several Omicron sublineages in relatively low numbers during our surveillance, in accordance with the low frequency of the mutation. In detail, a study suggested that W152R, F157L, I210V, G257S, and D339H mutations slightly alter neutralization titers of sera against BA.2 lineages. The K147E mutation has exhibited a significant impact on polyclonal sera, but no effect on mAbs targeting RBD [84]. Due to the absence of these mutations in recent lineages, we can speculate that this mutation is not advantageous for the virus.

R158G mutation was first reported in May 2023 with low frequency, which increased in December 2024, following the trend of emerging mutations, as further national data suggest. In April 2024, this mutation disappeared in our cohort and is therefore considered a transient mutation. This mutation was already present in the Delta variant (June 2021–January 2022), linked to the fixed mutation G142D, which correlated to weaker interactions with antibodies and increased viral load [45].

Q183E (November 2022–February 2024) is an NTD mutation that had high prevalence in the XBB.1 and HV.1 sublineages. This substitution is associated with immune evasion in the XBB lineages, as it disrupts the hydrogen bond in the mAb C1520 in residue A32, causing a steric clash with residue W91, likely abrogating the binding of this specific mAb [85].

V213E mutation (in the NTD region), associated with lineage XBB.1.5, was found with high prevalence from November 2022 to February 2024. Conversely, the D215E mutation emerged from March to July 2022. Both mutations have been reported to not affect S protein structure and functions. G252V is an NTD mutation present with high frequency in XBB.1.5 and JG.3, which appeared in our study from November 2022 to January 2024. This mutation changes the NTD structure, resulting in decreased antibody recognition in lineages XBB.1.5, XBB.1.16, and EG.5 [43].

RBD mutations R346T and K444T were found between July 2022 and February 2024, and V445P between November 2022 and February 2024. These mutations altered the binding of BQ.1.1 to the RBD-ACE2 interface, which resulted in an increase in immune evasion capacity if compared with its ancestral state [85,86,87].

The L368I mutation, which emerged between November 2022 and February 2024, is associated with an increase in ACE2 binding affinity in lineages XBB.1.5 and BQ.1.1 [81].

The L452R mutation appeared in May 2022, but decreased drastically after February 2023 and maintained a low frequency until it disappeared in February 2024. Since December 2021, this mutation has been observed in Eastern regions, mainly in Indian sublineages and correlated with evasion from HLA-A24, thereby increasing replication capacity and promoting viral replication [88].

As previously mentioned, the FLip mutation L455F emerged from August 2023 until February 2024 in the JG.3 and HK.3 lineages. This mutation allows the virus to escape immune detection, although it remains sensitive to class 1 antibodies. The synergy between L455F and F456L results in a significant increase in ACE2 affinity [53]. As reported in the literature, FLip sublineages have gained the A475V mutation previously seen in a few BA.2.75 descendants, thus conferring evasion to class 1 antibodies [53]. This mutation emerged from September 2023 to January 2024. However, FLip’s variants suffered from the competition with the rapidly emerging JN.1 sublineages and disappeared in the first months of 2024.

The T478R mutation was found between April 2023 and January 2024 and is related to Omicron sublineages XBB.1.16, which has emerged in India and other Asian countries since early 2023. Due to its high transmissibility and immune evasion, this variant has been extensively investigated. In particular, the replacement of positively charged amino acids with non-polar amino acids results in an increase in the capacity to evade the immune system [85,89].

E484A was present for nearly two years, from March 2022 to February 2024, before it rapidly decreased. This mutation is associated with an overall decrease in SARS-CoV-2 fitness caused by a lower ACE2 binding capacity, although it favors antibody neutralization and results in a slightly better immune escape effect [57].

In the same residue, the F486V and F486S mutations emerged between June 2022 and July 2023 and between November 2022 and November 2023, respectively. These mutations are both associated with a significant ACE2 binding affinity decrease. Therefore, this suggests that a substitution at F486 leads to attenuated ACE2 binding affinity, although it contributes to neutralizing antibody evasion in both mutations. This is further supported by the F490P mutation observed from November 2023 to January 2024, which is found in sublineages of KC.1, which maintains this trait [90].

On the same residue, the F490S mutation persisted for an extended period, first emerging in November 2022 and remaining detectable until February 2024, reaching its peak prevalence between April and August 2023. This mutation is known to confer resistance to Bamlanivimab and Cilgavimab [91].

The P521S mutation was observed with a low frequency from March 2023 to December 2023. The low prevalence of this substitution seems to be justified, since it reduces viral infectivity without significantly impacting immune evasion [92,93].

The Q613H mutation was discovered infrequently and exclusively between February and June 2023. However, a study reports how this substitution has positive effects on SARS-CoV-2, as, together with the fixed mutations D614G and H655Y, it stabilizes S protein on virions [94].

The N658S mutation, located in the CTD2 of the S protein, emerged between July and October 2022. This mutation is rare and kept an overall low frequency in BA.4 lineages, which has enhanced antibody neutralization. However, studies suggest that the effect is not caused by N658S but by alteration in the saline bridge between RBD and class 3 antibodies, due to R346T/S/I mutations [95].

The A701V mutation, located on the S1/S2 cleavage site, was observed from May 2023 to January 2024. This substitution was previously discovered in Beta variant, and it is likely to hinder the attachment of S protein to ACE2, resulting in less virulent behavior but improved transmission. However, the emergence of new variants obscured those with A701V.

The S704L mutation was detected in JG.3 lineages between September 2023 and January 2024. This substitution was previously observed in Omicron subvariants, which caused a decrease in the surface expression of S protein. As a result, this mutation was not able to persist in viral populations over time [96]. All transient mutations identified in this study are listed in Table 4, and their spatial disposition is shown in Figure 11.

### 2.5. Divergent Mutations

The mutations that cannot be included in other categories due to their trend are included in the divergent mutation group and are displayed in Figure 12.

The L5F mutation was observed sporadically at low prevalence throughout our study period, exceeding a 1% frequency only during March and April 2022; from August to November 2022; in January, March, and June 2023; and from September to October 2023.

According to CoV-Spectrum, the trend of this substitution had some peaks throughout 2021, and since then has stabilized itself around 1% of sequences globally [97].

Studies suggest that the L5F mutation is one of many substitutions (S13I, L5F-S13I) that have a role in nuclear localization, increasing the hydrophobicity of the signal sequence of the S protein, facilitating entry into the cell endoplasmic reticulum for folding and assembly [98].

The F157S mutation appeared at low prevalence in June 2022 and disappeared by August 2022. It re-emerged with significantly higher prevalence in December 2023 and disappeared again by February 2024. This nonsynonymous mutation was identified as a destabilizing factor in S protein. Previous studies on molecular docking and binding free energy analyses suggest that the F157S mutation may reduce the binding affinity between the S protein and ACE2 receptor, contributing to variations in virulence and transmissibility [99].

The L212I mutation is a peculiar mutation that occurs when the first amino acid is deleted and the second is substituted in the 211–212 codons. This mutation was observed with low prevalence between March and June 2022 and then disappeared, until its prevalence rose drastically in November 2023. According to a recent study, these changes are linked to a decreased ability of peptides to bind to HLA-DRB1*03:01 [100].

A high prevalence of the V213G mutation was observed at the beginning of the study. Starting from December 2022, its prevalence gradually decreased, nearly disappearing in August 2023. In February 2024, mutation prevalence reached levels comparable to those observed at the beginning of the study, after the increase from October 2023 to December 2023. The literature lacks in-depth studies that could link this mutation to a specific effect. According to Sun et al. [101], this mutation is among the 15 mutations that are associated with increased infectivity and immune evasion.

The K356T mutation is responsible for the structural difference between XBB.1.5 and BA.2.86. This mutation appeared with low frequency in February, March, June, and August of 2023 but began to rise only in November of the same year. According to further in vitro analysis, this mutation is the cause of BA.2.86’s increased infectivity and immune evasion in comparison to XBB.1.5 [42].

There is a lack of information about D253G mutation, which was discovered in late 2020 in the USA, and it is related to the Iota variant. This mutation was observed in our study in September 2022, between April and May 2023, and lastly from September to December of the same year. The literature suggests that this mutation leads to immune evasion from antibodies targeting the NTD [102].

The G446S mutation reached high prevalence in April 2022 and then disappeared by June 2022. It reappeared at low prevalence in December 2022; then gradually increased, achieving high prevalence by April 2023; and remained stable until the end of our study period. The G446S mutation could lead to a decrease in viral fitness, causing further suppression in viral replication, as suggested by previous studies, due to the enhancement of T cell recognition [103,104].

The S939F mutation is closely related to BA.2.86 and JN.1 subvariants. In our study, it was only observed in November–December 2022 and, in higher prevalence, in November 2023. By modulating T-cell activation, this mutation could affect the effectiveness of the immune response [67,105].

All of these mutations have a divergent trend, appearing and disappearing more than once, at different frequencies over the period of our survey. These mutations, some more than others, undergo the so-called “Lazarus effect”—a concept originally used in paleobiology to describe so-called *Lazarus taxa*, i.e., species that reappear in the fossil record after a period during which they were thought to be extinct. In this virological context, this term is used by analogy to refer to mutations that seemingly vanish from the viral population only to re-emerge after several months in newly observed lineages. This apparent disappearance and resurgence can result from undetected low-frequency circulation, reintroduction from unsampled reservoirs, or new selective advantages in another context and time. This trend can be clearly seen in V213G mutation. We can try to explain this phenomenon, considering that mutations, which appear to be under negative selective pressure at first, may become advantageous later, in association with other variations. All divergent mutations identified in our study are listed in Table 5, and their spatial disposition is shown in Figure 13.

Functional analysis of the 88 identified mutations reveals that immune evasion represents the most common adaptive trait, with 56.8% (49 out of 88) of the mutations associated with increased escape from host immunity. This pattern is particularly noticeable in emerging and transient categories, which together account for 70% (35 out of 50) of immune-evasive mutations. The divergent group, although smaller, shows a similar trend, with six out of eight mutations enhancing immune evasion. Only one divergent mutation (G446S) is associated with reduced immune evasion. These findings suggest that immune escape undergoes a dominant selective pressure.

By comparison, mutations affecting ACE2 binding affinity represent the second-most frequent involvement, with 12 mutations linked to increased binding—primarily within the emerging group with 6 mutations—and 9 associated with reduced binding, mostly among transient mutations (5 mutations). Interestingly, six of the mutations that reduce ACE2 affinity co-occur with increased immune evasion and one with increased transmissibility, as represented in Figure 14. This may imply that immune escape could be prioritized over receptor binding optimization in certain evolutionary contexts. The list of these mutations is reported in Appendix A.

We analyzed potential O-glycosylation sites among mutations in the different categories at both the beginning and end of the study period. Excluding fixed mutations, as they are considered stably acquired, we observed that the loss of possible O-glycosylation sites was balanced by the appearance of new ones. Specifically, one site (T19I) was present only at the beginning, while at the end of the study, one site was lost due to the K356T substitution, but two new possible sites appeared (G446S and S939F). Overall, the total number of potential O-glycosylation sites remained stable over time.

## 3. Materials and Methods

### 3.1. Sample Collection and RNA Extraction

Viral RNA from 1333 nasopharyngeal swab samples (Copan, Brescia, Italy) collected from SARS-CoV-2 positive medical personnel and patients accessing the AOUI Hospital of Verona (Italy) was extracted as previously described [106,107].

RNA was extracted following the semi-automated protocol of the manufacturer’s instrument (Nimbus, Seegene, Seoul, Republic of Korea). Quantitative RT-PCR was performed on a Bio-Rad CFX 96 System (Hercules, CA, USA), using the Allplex SARS-CoV-2 Assay following the manufacturer’s instructions (Seegene, Seoul, Republic of Korea). Only samples with a cycle threshold (Ct) below 32 were taken into consideration for the study. For this reason, the choice of the sequencing samples was not totally random. This technical limitation could not be overcome since a lower Ct value implies a minor viral load and, consequently, suboptimal sequencing quality. Therefore, we sequenced not all but rather a portion of all SARS-CoV-2-positive samples from the AOUI Hospital of Verona.

### 3.2. Library Preparation and Sequencing

The library was prepared with an Illumina COVIDSeq Assay (Illumina, San Diego, CA, USA) as previously reported [107,108]. First, reverse transcription was performed on whole viral genome for each sample. Then, amplification was performed using ARTIC (Artic network) primer pool v4 and v4.1. Samples were sequenced with the Illumina MiSeq instrument (Illumina) in paired-end mode (2 × 150 bp) with V3 chemistry.

### 3.3. Bioinformatic Analysis

The sequence was analyzed by running a custom pipeline using SAMtools version 1.21 [109] and Minimap2 version 2.28 [110] on the Linux command line. Specifically, parameters for minimum depth were set to 50, the parameters for minimum mapping quality were set to 30, and the maximum call fraction was set to 0.9. Any base below these constraints was not called. The Pangolin COVID-19 Lineage Assigner v 3.0 [111] and Nextclade tool from Nextstrain (https://clades.nextstrain.org/ accessed on 10 January 2025) [112] were used to identify mutations and lineages. Further control of the distribution of sample reads was achieved using the Integrative Genomics Viewer (IGV) tool [113] and FastQC v 0.12.0. The SARS-CoV-2 reference genome (NC_045512.2) was used for the alignment. All sequences were then submitted to the national platform Italian COVID-19 Genomic (I-Co-Gen). Here, SARS-CoV-2 RECoVERY software v4.0 developed by the Istituto Superiore di Sanità (ISS) automatically performed another data quality control. Finally, all SARS-CoV-2 sequences were submitted to GISAID (Global Initiative on Sharing Avian Influenza Data) (https://gisaid.org/ accessed on 10 January 2025) [114] in the context of the Italian national surveillance project. Mutational analysis was curated manually based on literature reviews. Statistical analysis was conducted with Python v 3.8.10pandas module and visualized with Seaborn v 0.13.2 and Matplotlib v 3.10.5. Global prevalence data were taken from CoV-Spectrum (https://cov-spectrum.org accessed on 15 January 2025) [97].All 3D models were created using ChimeraX v 1.10.1 [115] using model 7A97 from the Protein Data Bank (PDB) (https://www.ebi.ac.uk/pdbe/ accessed on 20 January 2025).

## 4. Conclusions

Throughout our two-year epidemiological surveillance in Verona, we thoroughly examined SARS-CoV-2 spike protein mutations through the next generation sequencing (NGS) of COVID-19-positive nasopharyngeal swab samples. Our investigation revealed a significant number of mutations that are frequently linked to Omicron subvariants. The identified mutations had different effects on transmissibility, antibody resistance, and infectivity of viruses [9,10,11,62,101]. However, correlating individual mutations with specific effects was a challenge in certain cases due to the lack of information in the literature for certain aminoacidic changes. In several instances, functional implications were largely inferred from the literature and have not been experimentally validated yet. This highlights the need for future research incorporating experimental approaches, such as in vitro mutagenesis studies and in vivo infection models, to directly assess the biological consequences of mutations.

Our results represent a dynamic mutational landscape in which immune evasion is a major driver, which seems to be prioritized over receptor binding optimization. Furthermore, the overall number of potential O-glycosylation sites tends to remain consistent over time. Our findings are in line with mutational patterns reported globally, and the local mutation trend in a relatively small setting mirrors the trend of the wide-scale dynamics of the epidemic. Despite these considerations, we acknowledge that several limitations should be considered when interpreting our results. This was a single-center study, which may have restricted the generalizability of the findings compared with broader or more diverse populations. Moreover, while we normalized the data by month to account for variability in sample numbers over time, this approach may have still introduced bias when considering months with a low sampling density. Specifically, mandatory medical surveillance in the AOUI Hospital of Verona, which involved the screening of medical personnel for SARS-CoV-2, ended at the end of March 2023—halfway through the study. The reduction in the number of samples processed following the change in surveillance protocol could have impacted the statistical depth of the study. Lastly, the lack of patient demographic data does not permit the correlation between viral variants and host factors such as age, sex, or comorbidities. To validate and expand upon our observations, more multi-center studies are necessary, including diverse cohorts and comprehensive metadata.

Despite these limitations, our findings contribute valuable insights to the ongoing monitoring of SARS-CoV-2 evolution. Continuous genomic surveillance, within a collaborative international framework, remains essential for the early detection of variants of concern and for the timely adaptation of vaccines and therapeutics [116,117]. In summary, this study underscores the importance of sustained, integrative surveillance to anticipate and mitigate the impact of emerging SARS-CoV-2 variants.

## Figures and Tables

**Figure 1 ijms-26-07558-f001:**
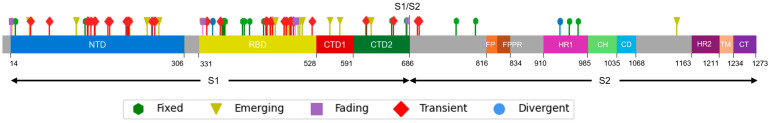
Mutation distribution along the S protein. Each mutation is colored differently according to the following classification: green, fixed; yellow, emerging; violet, fading; red, transient; and blue, divergent mutations. S1: Subunit 1; S2: Subunit 2; NTD: N-Terminal Domain; RBD: Receptor Binding Domain; CTD1: C-terminal domain 1; and CTD2: C-terminal domain 2; FP: fusion peptide; FPPR: fusion-peptide proximal region; HR1: heptad repeat 1; CH: central helix; CD: connector domain; HR2: heptad repeat 2; TM: transmembrane segment; CT: cytoplasmic tail.

**Figure 2 ijms-26-07558-f002:**
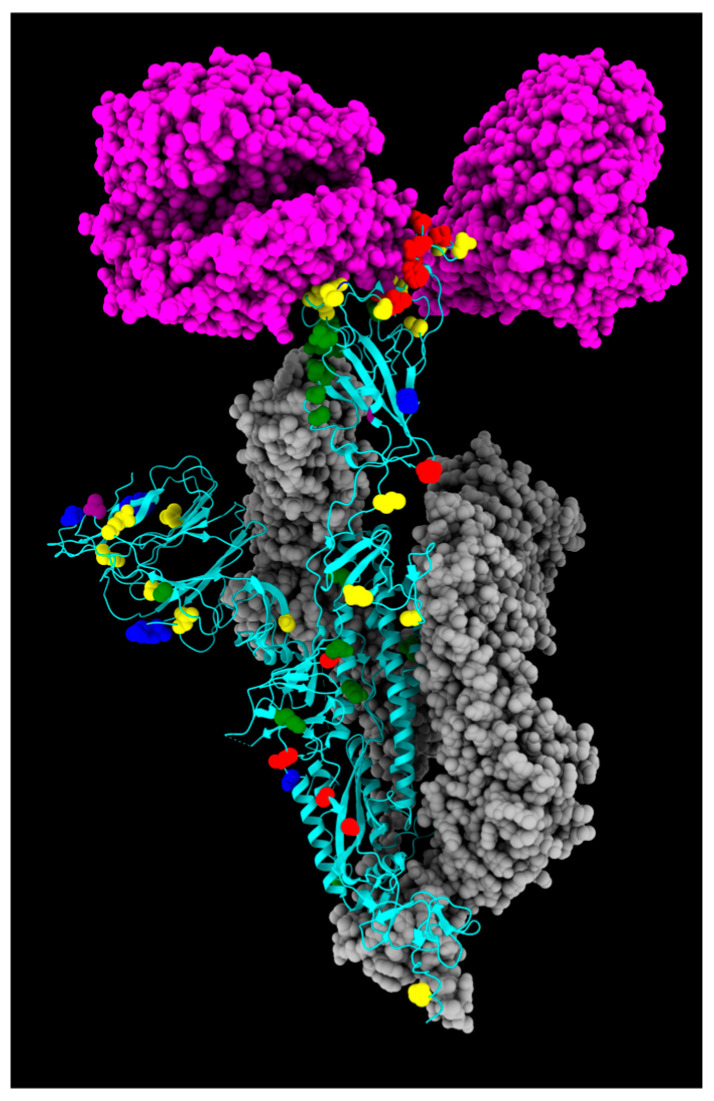
Three-dimensional structure of spike protein, with all mutations described in this study highlighted in different colors: green, fixed; yellow, emerging; violet, fading; red, transient; and blue, divergent mutations. Cyan, gray, and magenta are used to represent one chain of the protein, the other spike chains of the trimer, and the ACE2 receptor, respectively.

**Figure 3 ijms-26-07558-f003:**
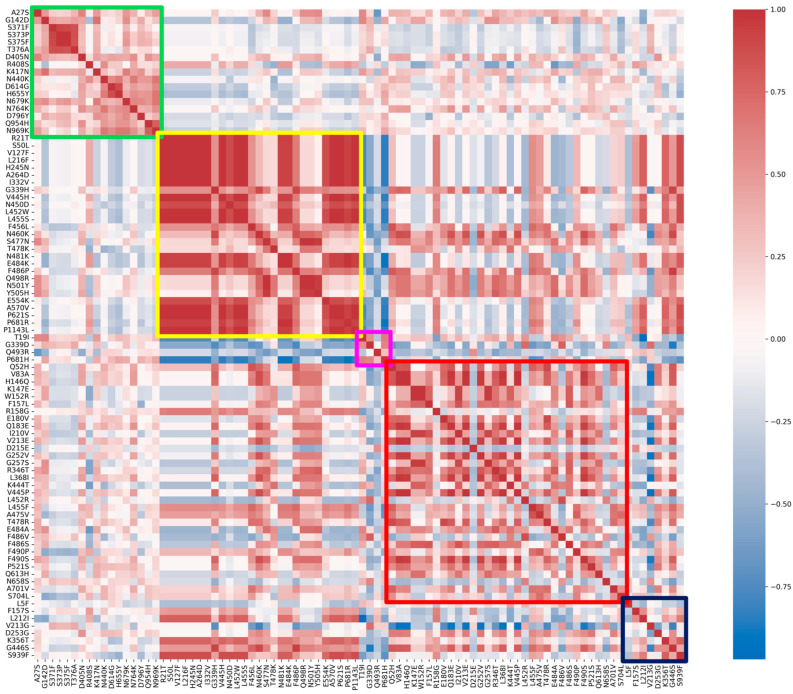
Spearman’s correlation heatmap matrix for all substitutions, grouped by the classification given in this study. The colored boxes highlight the specific correlations between substitutions belonging to the same group: green, fixed; yellow, emerging; violet, fading; red, transient; and blue, divergent mutations.

**Figure 4 ijms-26-07558-f004:**
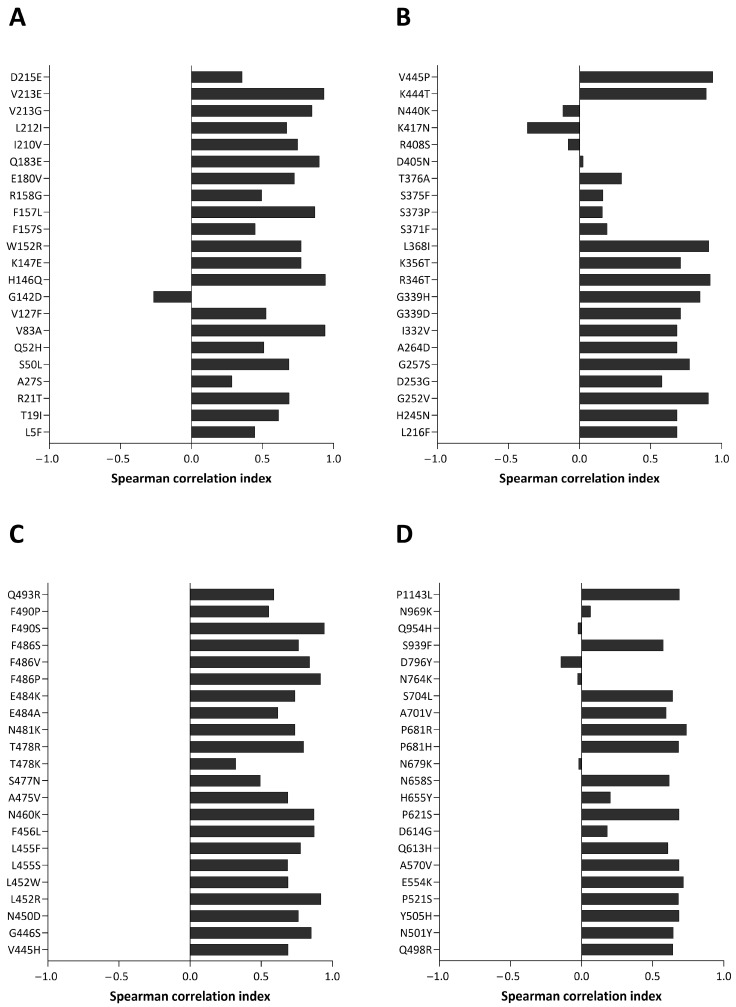
Plot of Spearman’s correlation index of the global trend compared to that observed in this study for each mutation. Almost all mutations display a positive correlation index, indicating that the trend of substitution considered in this study mirrors the global one. Panels (**A**–**D**) each show 22 substitutions, ordered according to their amino acid position along the S protein.

**Figure 5 ijms-26-07558-f005:**
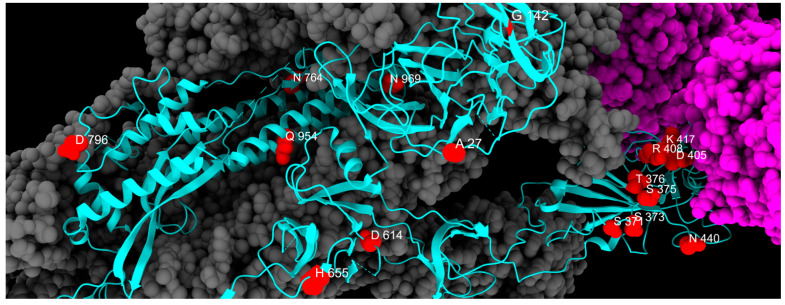
Distribution of fixed mutation sites, colored in red, along one chain of S protein, colored in cyan. Gray and magenta represent the other spike chains of the trimer and the ACE2 receptor, respectively.

**Figure 6 ijms-26-07558-f006:**
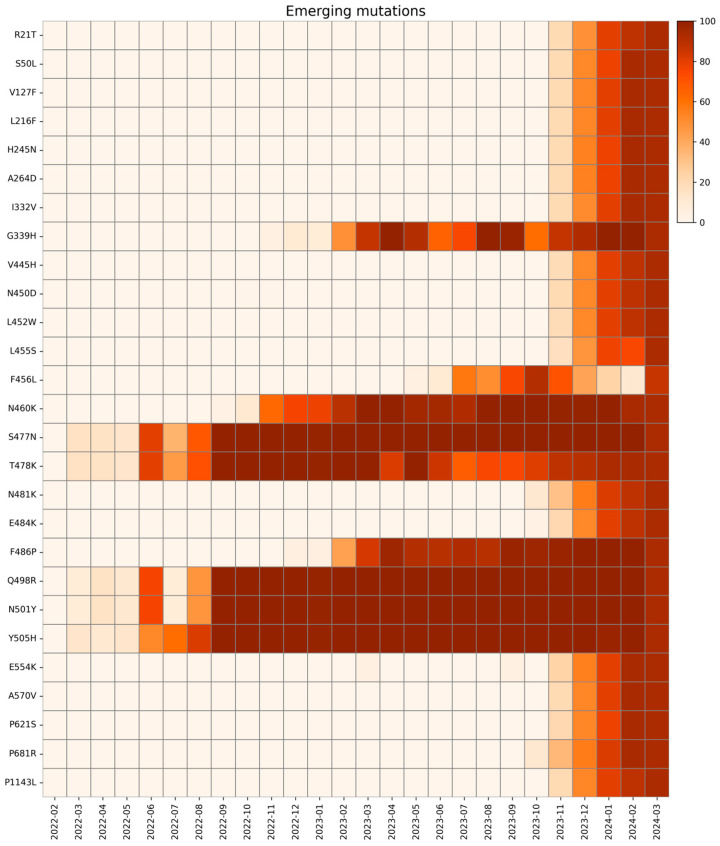
Heatmap with the relative monthly prevalence of the emerging mutations in the period from February 2022 to March 2024.

**Figure 7 ijms-26-07558-f007:**
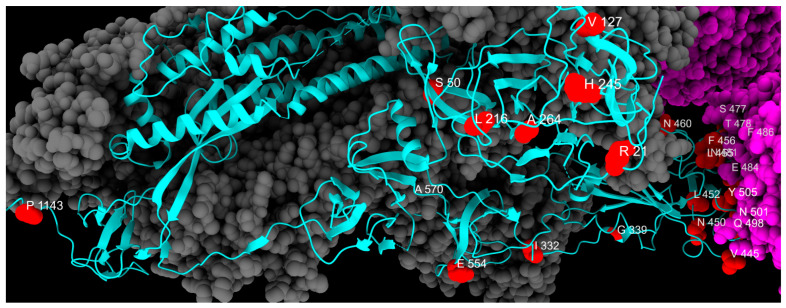
Distribution of emerging mutation sites, colored in red, along one chain of S protein, colored in cyan. Gray and magenta represent the other spike chains of the trimer and the ACE2 receptor, respectively.

**Figure 8 ijms-26-07558-f008:**
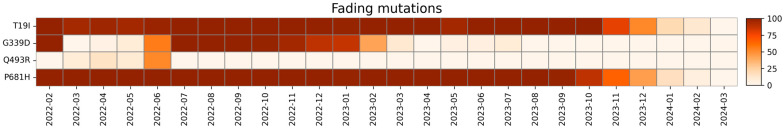
Heatmap with the relative monthly prevalence of the fading mutations in the period from February 2022 to March 2024.

**Figure 9 ijms-26-07558-f009:**
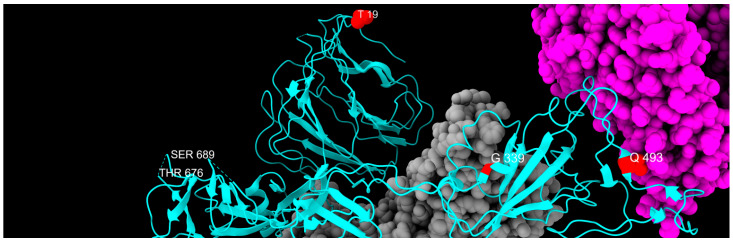
Distribution of fading mutation sites, colored in red, along one chain of S protein, colored in cyan, with the exception of P681H, which is absent from the PDB model and is limited to a side loop. Gray and magenta represent the other spike chains of the trimer and the ACE2 receptor, respectively.

**Figure 10 ijms-26-07558-f010:**
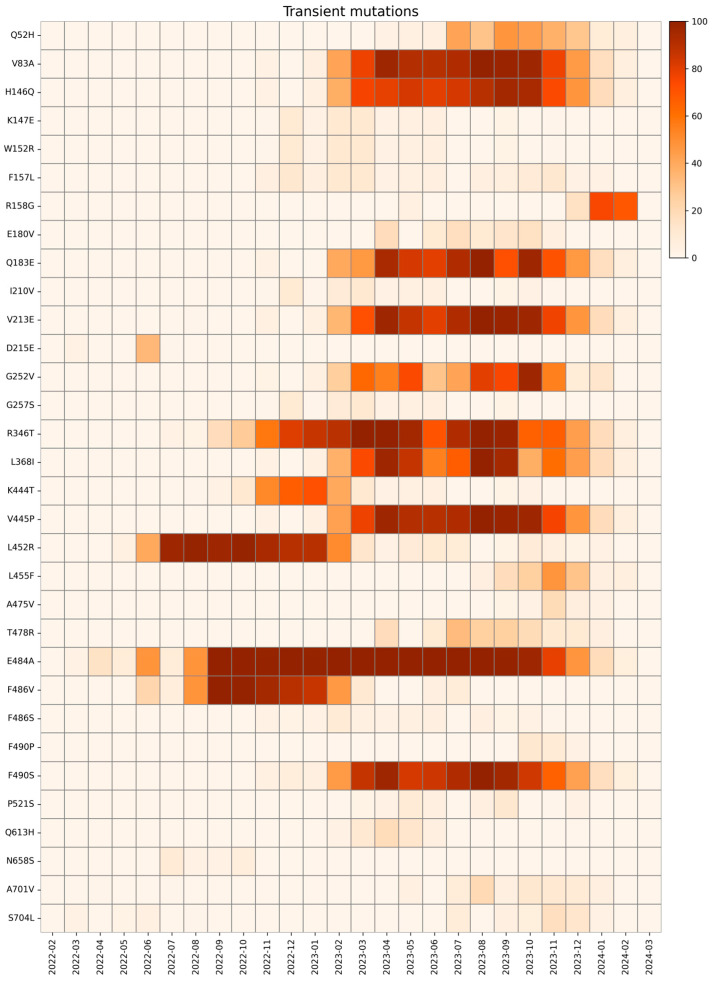
Heatmap with the relative monthly prevalence of transient mutations in the period from February 2022 to March 2024.

**Figure 11 ijms-26-07558-f011:**
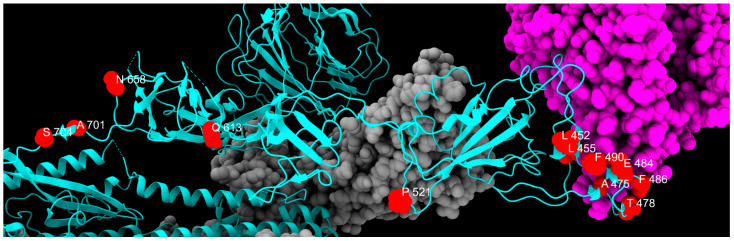
Distribution of transient mutation sites, colored in red, along one chain of S protein, colored in cyan. Gray and magenta represent the other spike chains of the trimer and the ACE2 receptor, respectively.

**Figure 12 ijms-26-07558-f012:**
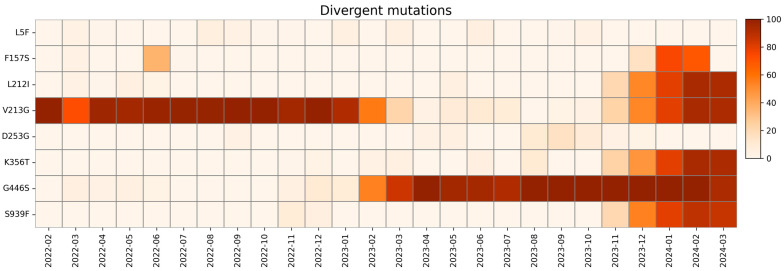
Heatmap with relative monthly prevalence of divergent mutations in the period from February 2022 to March 2024.

**Figure 13 ijms-26-07558-f013:**
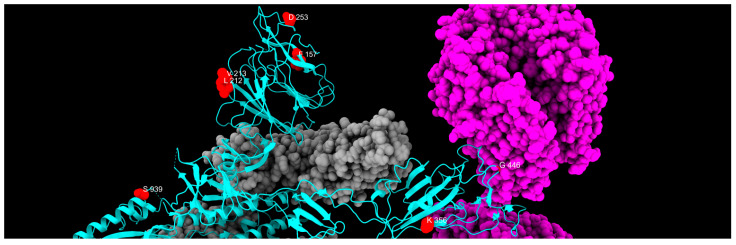
Distribution of divergent mutation sites, colored in red, along one chain of the S protein, colored in cyan. Gray and magenta represent the other spike chains of the trimer and the ACE2 receptor, respectively.

**Figure 14 ijms-26-07558-f014:**
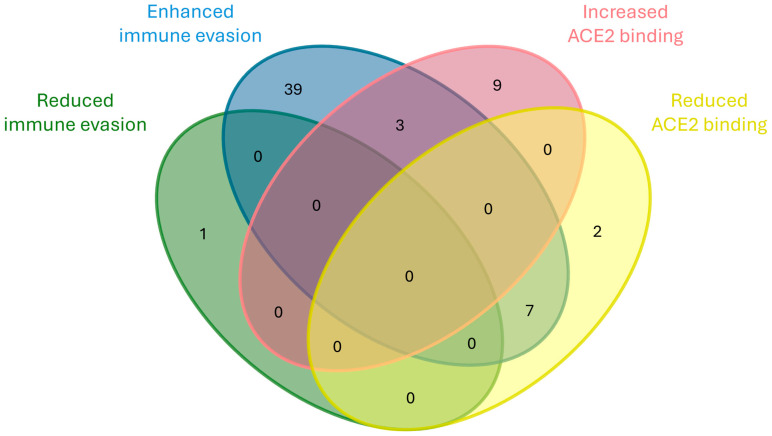
Venn diagram showing the co-occurrence of mutation involved, either positively or negatively, in immune evasion and ACE2 binding.

**Table 1 ijms-26-07558-t001:** List and features of the 17 identified fixed mutations. The table reports the change in the polarity of the residue involved in mutations, as well as their domain location and functional involvement. The year and month in which the prevalence rate of each mutation first exceeded 90% globally are reported.

Fixed Mutations	Wild Type Polarity	Mutated Polarity	Domain	Involvement	First > 90% Prevalence Date
A27S	Hydrophobic	Polar	NTD	Immune evasion	April 2022
G142D	Glycine	Negative	NTD	Immune evasion and increased viral loads	April 2022
S371F	Polar	Hydrophobic	RBD	Cell entry	April 2022
S373P	Polar	Proline	RBD	Increased ACE2 binding	February 2022
S375F	Polar	Hydrophobic	RBD	Reduced pathogenicity	February 2022
T376A	Polar	Hydrophobic	RBD	Reduced pathogenicity	April 2022
D405N	Negative	Polar	RBD	Immune evasion	April 2022
R408S	Positive	Polar	RBD	Immune evasion and increased ACE2 binding	April 2022
K417N	Positive	Polar	RBD	Immune evasion and reduced ACE2 binding	March 2022
N440K	Polar	Positive	RBD	Immune evasion and reduced ACE2 binding	September 2022
D614G	Negative	Glycine	CTD2	Increased ACE2 binding and increased replication	April 2020
H655Y	Positive	Hydrophobic	CTD2	Increased endosomal entry	January 2022
N679K	Polar	Positive	CTD2	Possible increased furin-mediated cleavage	January 2022
N764K	Polar	Positive	S2 cleavage site	Possible new cleavage site introduction	January 2022
D796Y	Negative	Hydrophobic	S2 cleavage site	Immune evasion	January 2022
Q954H	Polar	Positive	HR1	Immune evasion	January 2022
N969K	Polar	Positive	HR1	Increased endosomal entry	January 2022

**Table 2 ijms-26-07558-t002:** List and features of identified emerging mutations. The table reports the change in the polarity of the residue involved in mutations, as well as their domain location and functional involvement.

Emerging Mutations	Wild Type Polarity	Mutated Polarity	Domain	Involvement
R21T	Positive	Polar	NTD	Unknown
S50L	Polar	Hydrophobic	NTD	Increased lung cell entry
V127F	Hydrophobic	Hydrophobic	NTD	Immune evasion
L216F	Hydrophobic	Hydrophobic	NTD	Immune evasion
H245N	Positive	Polar	NTD	Immune evasion
A264D	Hydrophobic	Negative	NTD	Immune evasion
I332V	Hydrophobic	Hydrophobic	RBD	Immune evasion
G339H	Glycine	Positive	RBD	Immune evasion
V445H	Hydrophobic	Positive	RBD	Immune evasion
N450D	Polar	Negative	RBD	Immune evasion
L452W	Hydrophobic	Hydrophobic	RBD	Immune evasion
L455S	Hydrophobic	Polar	RBD	Immune evasion and reduced ACE2 binding
F456L	Hydrophobic	Hydrophobic	RBD	Increased ACE2 binding with L455F
N460K	Polar	Positive	RBD	Immune evasion
S477N	Polar	Polar	RBD	Immune evasion and increased ACE2 binding
T478K	Polar	Positive	RBD	Increased ACE2 binding
N481K	Polar	Positive	RBD	Immune evasion
E484K	Negative	Positive	RBD	Immune evasion
F486P	Hydrophobic	Proline	RBD	Immune evasion
Q498R	Polar	Positive	RBD	Increased ACE2 binding
N501Y	Polar	Hydrophobic	RBD	Immune evasion and increased ACE2 binding
Y505H	Hydrophobic	Positive	RBD	Increased ACE2 binding
E554K	Negative	Positive	CTD1	Immune evasion
A570V	Hydrophobic	Hydrophobic	CTD1	Trimer stability
P621S	Proline	Polar	CTD2	Reduced fusogenicity
P681R	Proline	Positive	CTD2	Increased pathogenicity
P1143L	Proline	Hydrophobic	S2-CTD	Possible increased cell entry

**Table 3 ijms-26-07558-t003:** List and features of identified fading mutations. The table reports the change in the polarity of the residue involved in mutations, as well as their domain location and functional involvement.

Fading Mutations	Wild Type Polarity	Mutated Polarity	Domain	Involvement
T19I	Polar	Hydrophobic	NTD	Reduced fusogenicity
G339D	Glycine	Negative	RBD	Immune evasion
Q439R	Polar	Positive	RBD	Increased ACE2 binding
P681H	Proline	Positive	CTD2	Increased furin-mediated cleavage

**Table 4 ijms-26-07558-t004:** List and features of identified transient mutations. The table reports the change in the polarity of the residue involved in mutations, as well as their domain location and functional involvement.

Transient Mutations	Wild Type Polarity	Mutated Polarity	Domain	Involvement
Q52H	Polar	Positive	NTD	Unknown
V83A	Hydrophobic	Hydrophobic	NTD	Increased fusogenicity
H146Q	Positive	Polar	NTD	Unknown
K147E	Positive	Negative	NTD	Immune evasion
W152R	Hydrophobic	Positive	NTD	Immune evasion
F157L	Hydrophobic	Hydrophobic	NTD	Immune evasion
R158G	Positive	Glycine	NTD	Immune evasion and increased viral load
E180V	Negative	Hydrophobic	NTD	Increased stability
Q183E	Polar	Negative	NTD	Immune evasion
I210V	Hydrophobic	Hydrophobic	NTD	Immune evasion
V213E	Hydrophobic	Negative	NTD	Unknown
D215E	Negative	Negative	NTD	Unknown
G252V	Glycine	Hydrophobic	NTD	Immune evasion
G257S	Glycine	Polar	NTD	Immune evasion
R346T	Positive	Polar	RBD	Increased fusogenicity
L368I	Hydrophobic	Hydrophobic	RBD	Increased ACE2 binding
K444T	Positive	Polar	RBD	Immune evasion
V445P	Hydrophobic	Proline	RBD	Immune evasion
L452R	Hydrophobic	Positive	RBD	Immune evasion
L455F	Hydrophobic	Hydrophobic	RBD	Increased ACE2 binding with F456L
A475V	Hydrophobic	Hydrophobic	RBD	Immune evasion
T478R	Polar	Positive	RBD	Immune evasion
E484A	Negative	Hydrophobic	RBD	Immune evasion and reduced ACE2 binding
F486V	Hydrophobic	Hydrophobic	RBD	Immune evasion and reduced ACE2 binding
F486S	Hydrophobic	Polar	RBD	Immune evasion and reduced ACE2 binding
F490P	Hydrophobic	Proline	RBD	Immune evasion and reduced ACE2 binding
F490S	Hydrophobic	Polar	RBD	Immune evasion
P521S	Proline	Polar	RBD	Reduced viral infectivity
Q613H	Polar	Positive	CTD2	Protein stabilization
N658S	Polar	Polar	CTD2	Unknown
A701V	Hydrophobic	Hydrophobic	S2 cleavage site	Reduced ACE2 binding
S704L	Polar	Hydrophobic	S2 cleavage site	Reduced surface expression

**Table 5 ijms-26-07558-t005:** List and features of identified divergent mutations. The table reports the change in the polarity of the residue involved in mutations, as well as their domain location and functional involvement.

Divergent Mutations	Wild Type Polarity	Mutated Polarity	Domain	Involvement
L5F	Hydrophobic	Hydrophobic	NTD	Enhanced ER targeting and protein processing
F157S	Hydrophobic	Hydrophobic	NTD	Increased transmissibility and reduced ACE2 binding
L212I	Hydrophobic	Hydrophobic	NTD	Immune evasion
V213G	Hydrophobic	Glycine	NTD	Immune evasion and increased infectivity
D253G	Negative	Glycine	NTD	Immune evasion
K356T	Positive	Polar	RBD	Immune evasion and increased infectivity
G446S	Glycine	Polar	RBD	Reduced immune evasion
S939F	Polar	Hydrophobic	HR1	Immune evasion

## Data Availability

The sequences analyzed in this study are openly available in GISAID at https://gisaid.org/ accessed on 20 January 2025.

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
