# Peer review of "An Italian Single-Center Genomic Surveillance Study: Two-Year Analysis of SARS-CoV-2 Spike Protein Mutations"

_ijms, 2025, doi:10.3390/ijms26157558_

Round 1
Reviewer 1 Report
Comments and Suggestions for Authors
This paper is merely a list of S protein mutations without any additional information of value and thus of limited interest
Some remarks:
Lines 37-39: “SARS-CoV-2 has a genome composed of about (not around) 30,000 nucleotides, encoding for spike (S), envelope (E), membrane (M), and nucleocapsid (N) structural proteins [1,2] and 16 non-structural proteins (NSP)” plus 7 accessory proteins!
Line 37: sars-Cov-2 : the name has to be given in full and its official one to be added (Betacoronavirus pandemicum).
Line 48-49: meaning? The RNA polymerase RNA dependant of the SARS-Cov-2 is highly stable and only a very few mutations concern its sequence. I do not see by which mechanism S1 may alter the activity of this polymerase.
As indicated by Andrew Rambaut et al. in Addendum to: Nat Microbiol 2020;5:1403–1407 https://doi.org/10.1038/s41564-020-0770-5, PANGOLIN should be used for the program that is limited to SARS-Cov-2. It is used by WHO for variant classification but not directly related to it
line 67: “increased SARS-CoV-2 viral loads” ? The references 8 and 9 are solely on affinity between S1 and ACE2 and infectivity of mutant D614G not working with live virus.
Lines 71 and 76: virulence ? How spike mutations can give clue on the unfolding of viral cycle?
Author Response
Comment 1: Lines 37-39: “SARS-CoV-2 has a genome composed of about (not around) 30,000 nucleotides, encoding for spike (S), envelope (E), membrane (M), and nucleocapsid (N) structural proteins [1,2] and 16 non-structural proteins (NSP)” plus 7 accessory proteins!
Response 1: We added a reference and modified as follows:
Lines 38-40: “SARS-CoV-2 has a genome composed of around about 30,000 nucleotides, encoding for spike (S), envelope (E), membrane (M), and nucleocapsid (N) structural proteins, 16 non-structural proteins (NSP) and seven confirmed accessory proteins [1-3].”
Comment 2: Line 37: sars-Cov-2 : the name has to be given in full and its official one to be added (Betacoronavirus pandemicum).
Response 2: We have modified accordingly at the first occurrence as follows:
Line 34-35: “The severe acute respiratory syndrome coronavirus 2 (SARS-CoV-2, Betacoronavirus pandemicum) virus, responsible…”
Comment 3: Line 48-49: meaning? The RNA polymerase RNA dependant of the SARS-Cov-2 is highly stable and only a very few mutations concern its sequence. I do not see by which mechanism S1 may alter the activity of this polymerase.
Response 3: This was a typo and we modified as follows:
Line 49-50: “…due to nucleotide substitutions, insertions, and deletions that affect the functionality of the RNA polymerase.”
Comment 4: As indicated by Andrew Rambaut et al. in Addendum to: Nat Microbiol 2020;5:1403–1407 https://doi.org/10.1038/s41564-020-0770-5, PANGOLIN should be used for the program that is limited to SARS-Cov-2. It is used by WHO for variant classification but not directly related to it
Response 4: If referring to line 52, we modified accordingly as follows:
Line 51-52 : “The WHO classified SARS-CoV-2 strains are classified in Phylogenetic Assignment of Named Global Outbreak (PANGO)…”
Comment 5: line 67: “increased SARS-CoV-2 viral loads” ? The references 8 and 9 are solely on affinity between S1 and ACE2 and infectivity of mutant D614G not working with live virus.
Response 5: Reference 8 (reference 9 after revisions, doi:10.1016/j.cell.2020.06.043) states “G614 is associated with potentially higher viral loads in COVID-19 patients but not with disease severity”.
Comment 6: Lines 71 and 76: virulence ? How spike mutations can give clue on the unfolding of viral cycle?
Response 6: According to virulence definition (doi: 10.1186/1741-7007-10-6), S mutations might increase its affinity binding with cell receptor and then potentially enhance viral entry and to favor the target selection (i.e. higher infection of lower or upper airways). Hence, even the impact of viral infection on cell biology may be increased following the S binding affinity variations by mutations. To rule out any misleading, we can delete the word virulence from sentence (line 73).

Reviewer 2 Report
Comments and Suggestions for Authors
This manuscript presents a two-year genomic surveillance study from Verona, Italy, analyzing mutations in the SARS-CoV-2 spike protein across 1333 samples collected between February 2022 and April 2024. The authors categorize spike protein mutations into five distinct groups: fixed, emerging, fading, transient, and divergent, and investigate their prevalence, temporal dynamics, structural locations, and potential functional implications. The study is relevant and comprehensive. There are extensive and appropriate citations to the literature. However, it reads somewhat like a catalog rather than an analysis that would make it a more significant contribution. I provide suggestions for substantial revisions below.
Major Revisions
The authors might consider updating the tables to include more useful details, or creating an additional summary table that highlights the most important or impactful mutations. For instance, Table 1 could include when each fixed mutation first appeared. Other tables mention mutations linked to immune evasion or vaccine resistance, but this information could be pulled together more clearly. Even if the tables can’t be expanded due to space, the narrative could be better organized. For example, mutations could be grouped by function or significance, which would help readers follow the findings more easily. These changes would make the manuscript clearer and increase its overall impact.
The definition of classifications must be made clearer and more quantitative (that is only the case for the transient) class since it is not based on a standard methodology (see ll. 107-114).
The only statistic that is considered here is frequency of the whole, which is of limited value since the samples were collected over time, and there are different numbers of samples at different time points, representing different sampling rates --- meaning that frequencies will be biased towards periods where there were higher sampling rates (hence mutations at those times would appear at greater frequency). The authors should address this issue. For example, The authors they could normalize mutation frequencies by the number of samples collected in each time period and/or provide monthly/quarterly sampling numbers. They could also consider using prevalence rates within time windows rather than overall frequency and apply time-series analysis approaches.
Additionally, there should be clearer identification of mutation co-occurrence patterns, particularly mutations that frequently appear together within the same lineages or across distinct sublineages. Highlighting such associations could offer insights into epistatic interactions or synergistic effects that drive variant fitness. The manuscript also lacks statistical analysis of mutational trends—such as correlation analysis, clustering, or co-occurrence frequency—that would support claims of evolutionary significance. Incorporating phylogenetic context or lineage-based mutation mapping would also strengthen interpretations, particularly where certain mutations appear recurrently or show convergent evolution across different variants. This would elevate the work beyond a descriptive catalogue to a more mechanistically informative analysis.
Graphical elements could also be added to improve readability and provide greater assistance to the reader. For instance, the authors could include graphics such as heatmaps, sankey diagrams, or other kinds of graphs to show the prevalence or incidence of mutations over time. This could also be keyed to the point at which the population in the geographical area of the study reached different thresholds of vaccination, or levels of disease prevalence. Another point is that while there is a domain mapping in Figure 1, it might be helpful to show key mutations on a 3D structure.
The Discussion should contextualize the manuscripts’ findings by comparing patterns observed in the study area with those reported from other regions, e.g., in Italy, Europe, and globally. Are the temporal dynamics similar? Are there region-specific patterns? This would help readers understand whether findings are generalizable or reflect local epidemiological factors.
The Methods must detail the sampling strategy: Were these consecutive samples? Random selection? From which hospitals/clinics? What were the inclusion/exclusion criteria? Include basic demographics (age distribution, sex, vaccination status) and clinical metadata (severity, hospitalization status). Explicitly discuss potential sampling biases and how they might affect the generalizability of findings
While there is some description of sequencing and analysis methods, it would be helpful to specify minimum sequencing depth, genome coverage thresholds for inclusion, quality score cutoffs, and variant calling parameters; also including, e.g., how low-frequency variants were handled and whether mutations were manually verified, especially for key findings. I understand that a lot of this is incorporated by reference to established techniques, but some of these parameters can vary and should be clarified in the manuscript itself. Once revised, the Methods section should be broken up into multiple subsections. The authors should consider also including metrics on sequencing success rate and reasons for sample exclusion.
The manuscript should acknowledge key limitations: (1) functional implications are largely inferred from literature rather than experimentally validated; (2) the single-center design limits generalizability; (3) lack of clinical outcome data prevents assessment of mutation impact on disease severity; (4) absence of immunological data (antibody levels, prior infection history) limits interpretation of immune escape patterns. Future directions should be proposed to address these gaps.
Minor Revisions
The brief discussion of the “Lazarus effect” does not fully make sense at ll. 457-459. It should be clarified for improved readability.
At ll. 179-182 there is a reference to “S216F,” which I think was meant to be L216F. Please carefully review to make sure there are no typographical errors. (See also, e.g., “D339H” at l. 316.)
The paragraph at ll. 214-250 is extremely long and should be broken up to improve readability.
Author Response
Reviewer 2:
Comment 1: The authors might consider updating the tables to include more useful details, or creating an additional summary table that highlights the most important or impactful mutations. For instance, Table 1 could include when each fixed mutation first appeared. Other tables mention mutations linked to immune evasion or vaccine resistance, but this information could be pulled together more clearly. Even if the tables can’t be expanded due to space, the narrative could be better organized. For example, mutations could be grouped by function or significance, which would help readers follow the findings more easily. These changes would make the manuscript clearer and increase its overall impact.
Response 1: Thank you for your valuable suggestion. We have updated Table 1 to include the date when the prevalence of each mutation first exceeded 90%, indicating the point at which it became globally fixed.
Regarding the idea of grouping mutations by function, we considered this approach but concluded that it could be confusing, as many mutations are involved in multiple functional aspects. To maintain clarity, we have included a brief summary of the known functional implications of each mutation within the relevant tables instead. We believe this format improves data visualization and makes the information more accessible to readers. Additionally, we removed the overall frequency column, as it may be less informative in the current context.
Comment 2: The definition of classifications must be made clearer and more quantitative (that is only the case for the transient) class since it is not based on a standard methodology (see ll. 107-114).
Response 2: We have modified accordingly as follows:
Lines 116-130: “Based on their dynamics, the spike mutations were classified into five distinct categories: fixed mutations (n = 17), consistently detected with stable prevalence throughout the study period (whose normalized monthly prevalence was equal to or greater than 90% throughout the entire observation period, allowing for no more than two consecutive months below 90%); emerging mutations (n = 27), initially absent but subsequently appearing and persisting over time (whose relative monthly prevalence was 0% for at least the first month of the study, followed by a sustained increase in relative monthly prevalence, eventually exceeding 90%) with an ascending trend; fading mutations (n = 4), present at the beginning of the surveillance but later disappearing (with a relative monthly prevalence above 1% for at least four consecutive months at the beginning of the surveillance, followed by a decline to 0%) with a descending trend; transient mutations (n = 32), characterized by a temporary rise in prevalence (above 1% for at least two consecutive months) followed by a sustained decline; and divergent mutations (n = 8), exhibiting fluctuations with multiple phases of appearance and disappearance that do not meet the criteria of any of the previous categories.”
Comment 3: The only statistic that is considered here is frequency of the whole, which is of limited value since the samples were collected over time, and there are different numbers of samples at different time points, representing different sampling rates --- meaning that frequencies will be biased towards periods where there were higher sampling rates (hence mutations at those times would appear at greater frequency). The authors should address this issue. For example, the authors they could normalize mutation frequencies by the number of samples collected in each time period and/or provide monthly/quarterly sampling numbers. They could also consider using prevalence rates within time windows rather than overall frequency and apply time-series analysis approaches.
Response 3: The data are already normalized by the number of samples collected each month, this point has been now clarified when defining the classifications (see Response 2).
Comment 4: Additionally, there should be clearer identification of mutation co-occurrence patterns, particularly mutations that frequently appear together within the same lineages or across distinct sublineages. Highlighting such associations could offer insights into epistatic interactions or synergistic effects that drive variant fitness. The manuscript also lacks statistical analysis of mutational trends—such as correlation analysis, clustering, or co-occurrence frequency—that would support claims of evolutionary significance. Incorporating phylogenetic context or lineage-based mutation mapping would also strengthen interpretations, particularly where certain mutations appear recurrently or show convergent evolution across different variants. This would elevate the work beyond a descriptive catalogue to a more mechanistically informative analysis.
Response 4: Addition of heatmaps to the manuscript may help visualize co-occurrence patterns of mutations. We also added a Spearman correlation heatmap to better visualize correlations between and within each group.
Lines 133-135: “Correlation between substitutions and, specifically, within each group, is visualized in the Spearman correlation heatmap in Figure 3. The fixed and emerging groups show higher correlation values when compared to the other three.”
Comment 5: Graphical elements could also be added to improve readability and provide greater assistance to the reader. For instance, the authors could include graphics such as heatmaps, sankey diagrams, or other kinds of graphs to show the prevalence or incidence of mutations over time. This could also be keyed to the point at which the population in the geographical area of the study reached different thresholds of vaccination, or levels of disease prevalence. Another point is that while there is a domain mapping in Figure 1, it might be helpful to show key mutations on a 3D structure.
Response 5: Heatmaps and 3D structure highlighting the mutations for each of the categories have been added to the manuscript.
Comment 6: The Discussion should contextualize the manuscripts’ findings by comparing patterns observed in the study area with those reported from other regions, e.g., in Italy, Europe, and globally. Are the temporal dynamics similar? Are there region-specific patterns? This would help readers understand whether findings are generalizable or reflect local epidemiological factors.
Response 6: All trends for the mutations reported in our study mirrors the global one, as seen from CoV-Spectrum. However, due to the large number of mutations taken in consideration, it would be unfeasible to show the comparison trend for each one.
We added this part in the discussion session, instead:
Lines 130-132: “The reported trends for the mutations described below have been manually verified on CoV-Spectrum, with each one mirroring the global temporal dynamics observed during the same period.”
Comment 7: The Methods must detail the sampling strategy: Were these consecutive samples? Random selection? From which hospitals/clinics? What were the inclusion/exclusion criteria? Include basic demographics (age distribution, sex, vaccination status) and clinical metadata (severity, hospitalization status). Explicitly discuss potential sampling biases and how they might affect the generalizability of findings
Response 7: The methods section was modified as follows:
Lines 555-567: “Viral RNA from 1333 nasopharyngeal swab samples (Copan, Brescia, Italy) collected from SARS-CoV-2 positive patients accessing the AOUI Hospital of Verona (Italy) was extracted as previously described[107,108].
RNA extraction was performed following the semi-automated protocol of the manufacturer instrument (Nimbus, Seegene). Quantitative RT-PCR was performed on Bio-Rad CFX 96 System (California, USA), using the Allplex SARS-CoV-2 Assay following manufacturer instructions (Seegene, Seoul, South Korea). Only samples having cycle threshold (Ct) below 32 were taken into consideration for the study.
For this reason, the choice of the sequencing samples is not totally random. This technical limitation could not be overcome since a lower Ct value implies a minor viral load and, consequently, suboptimal sequencing quality. Therefore, we sequenced not all but rather a portion of all SARS-CoV-2-positive samples in the AOUI Hospital of Verona.”
With regard to the basic demographics and the clinical metadata, we are sorry to say that we are unable to add this information as we do not have the permission.
Comment 8: While there is some description of sequencing and analysis methods, it would be helpful to specify minimum sequencing depth, genome coverage thresholds for inclusion, quality score cutoffs, and variant calling parameters; also including, e.g., how low-frequency variants were handled and whether mutations were manually verified, especially for key findings. I understand that a lot of this is incorporated by reference to established techniques, but some of these parameters can vary and should be clarified in the manuscript itself. Once revised, the Methods section should be broken up into multiple subsections. The authors should consider also including metrics on sequencing success rate and reasons for sample exclusion.
Response 8: We agree and divided the Methods in different paragraphs, adding this part:
Line 576-579: “Specifically, parameters for minimum depth were set to 50, the parameters for mini-mum mapping quality were set to 30 and the maximum call fraction was set to 0.9. Any base below these constraints was not called.”
Comment 9: The manuscript should acknowledge key limitations: (1) functional implications are largely inferred from literature rather than experimentally validated; (2) the single-center design limits generalizability; (3) lack of clinical outcome data prevents assessment of mutation impact on disease severity; (4) absence of immunological data (antibody levels, prior infection history) limits interpretation of immune escape patterns. Future directions should be proposed to address these gaps.
Response 9: Thank you for your suggestion. In regards of point (3) and (4): our goal was the study and comparison of SARS-CoV-2 sequences. Moreover, the samples were anonymized and it is impossible to retrieve all clinical and immunological data. Moreover, the reduced pathogenicity of the Omicron variant, to which all our data are referred, makes difficult to assess the real clinical significance of the infection, especially in pluri-pathological patients such as hospitalized one.
We modified the conclusion as follows:
Lines 592-615: “Throughout our two-year epidemiological surveillance in Verona, we performed a thorough examination of SARS-CoV-2 spike protein mutations by employing Next Generation Sequencing (NGS) of COVID 19-positive nasopharyngeal swab samples. Our investigation revealed a significant number of mutations, which are frequently linked to Omicron subvariants. The identified mutations had different effects on transmissibility, antibody resistance, and infectivity of viruses. However, correlating individual mutations with specific effects was a challenge in certain cases due to the lack of information in literature for certain aminoacidic changes. In several instances, functional implications are largely inferred from literature and have not been experimentally validated yet. This highlights the need for future research incorporating experimental approaches, such as in vitro mutagenesis studies and in vivo infection models, to directly assess the biological consequences of mutations. Our findings are in line with mutational patterns reported globally, and the local mutation trend in a relatively small reality, such as the AOUI of Verona, mirrors the trend of the wide-scale dynamics of the epidemic. Despite these considerations, we acknowledge that our single-center study may limit the generalizability of the results. More studies involving diverse cohorts will be necessary to validate and expand upon our observations.
Despite these limitations, our findings contribute valuable insights to the ongoing monitoring of SARS-CoV-2 evolution. Continuous genomic surveillance, within a collaborative international framework, remains essential for the early detection of variants of concern and for timely adaptation of vaccines and therapeutics [116,117]. In summary, this study underscores the importance of sustained, integrative surveillance to anticipate and mitigate the impact of emerging SARS-CoV-2 variants.”
Comment 10: The brief discussion of the “Lazarus effect” does not fully make sense at ll. 457-459. It should be clarified for improved readability.
Response 10: This section was modified as follows:
Lines 532-541: “These mutations, some more than others, undergo as the so-called “Lazarus effect” apparently disappearing and re-emerging in new lineages after several months — a concept originally used in paleobiology to describe so-called Lazarus taxa, i.e., species that reappear in the fossil record after a period during which they were thought to be extinct. In this virological context, this term is used by analogy to refer to mutations that seemingly vanish from the viral population, only to re-emerge after several months in newly observed lineages. This apparent disappearance and resurgence can result from undetected low-frequency circulation, reintroduction from unsampled reservoirs, or new selective advantages in another context and time.”
Comment 11: At ll. 179-182 there is a reference to “S216F,” which I think was meant to be L216F. Please carefully review to make sure there are no typographical errors. (See also, e.g., “D339H” at l. 316.)
Response 11: Thank you, we changed accordingly to “L216F”. Instead “D339H” is correct, as described in the nearby reference. The whole manuscript was checked and typos were corrected.
Comment 12: The paragraph at ll. 214-250 is extremely long and should be broken up to improve readability.
Response 12: The paragraph has been broken up.

Round 2
Reviewer 1 Report
Comments and Suggestions for Authors
The completion of the data by the authors render this paper acceptable for publication.
A little comment on the disappearance and appearance of Serine and Threonine in the S protein amino acid sequences, which are sites of O-glycosylation and thus play a role in virus/host relation, would have been welcome.
Author Response
Comment 1: The completion of the data by the authors render this paper acceptable for publication.
A little comment on the disappearance and appearance of Serine and Threonine in the S protein amino acid sequences, which are sites of O-glycosylation and thus play a role in virus/host relation, would have been welcome.
Response 1: We added the following parts in the Discussion and Conclusion sections, respectively:
Lines 576-582: “We analyzed potential O-glycosylation sites among mutations in the different categories at both the beginning and end of the study period. Excluding fixed mutations, as they are considered stably acquired, we observed that the loss of possible O-glycosylation sites was balanced by the appearance of new ones. Specifically, one site (T19I) was present only at the beginning, while at the end of the study, one site was lost due to the K356T substitution, but two new possible sites appeared (G446S and S939F). Overall, the total number of potential O-glycosylation sites remained stable over time.”
Lines 635-638: “Our results represent a dynamic mutational landscape in which immune evasion is a major driver, which seems to be prioritized over receptor binding optimization. Furthermore, the overall number of potential O-glycosylation sites tends to remain consistent over time.”

Reviewer 2 Report
Comments and Suggestions for Authors
I recognize and greatly appreciate the authors’ careful response to the comments in my previous review, and the many changes to the manuscript that they made in response to that review. I especially appreciate the additional graphical elements, which contribute significantly to readability and the impact of the paper. While the manuscript is greatly improved in many respects, I would recommend addressing the following issues to further strengthen the quality of the article:
1. While the authors have clarified their point about normalization by month and have explained the sampling that they did, they should address the limitations of doing that (and consolidate other study limitations such as the single study center point) in a dedicated section of the Conclusion or at the end of the Results and Discussion section). The authors could only note there that they did not have permission to report demographic data within that Limitations sub-section.
2. The added comment about global trends being mirrored is somewhat vague (lines 601-602). It would be helpful to have at least some exemplary discussion or clearer point (e.g., is there a lag? Are there continental differences? Are diversity patterns generally simiar?).
3. The authors should have at least some broader conclusions from their data that would help pull bring together the diverse results that are being presented and add analysis to what is currently still a somewhat descriptive overview of and help the user synthesize key observations. For instance, the authors could consider adding a summary figure or table that highlights a subset of particularly impactful mutations (e.g., those affecting immune escape, receptor binding, or showing convergent evolution). This would help orient readers amid a large volume of data.
4. There are still some language issues remaining in the paper. For example, in the revision in lines 600-603 describes Verona as a “relatively small reality.” Please make sure to review the material added in the revision to make sure that has been adequately edited.
With these additional improvements, I believe the manuscript would be better positioned for publication.
Author Response
Comment 1: While the authors have clarified their point about normalization by month and have explained the sampling that they did, they should address the limitations of doing that (and consolidate other study limitations such as the single study center point) in a dedicated section of the Conclusion or at the end of the Results and Discussion section). The authors could only note there that they did not have permission to report demographic data within that Limitations sub-section.
Response 1: We modified accordingly the Conclusion adding these considerations on the limitations of the study.
Lines 639-659: “Our findings are in line with mutational patterns reported globally, and the local mutation trend in a relatively small setting mirrors the trend of the wide-scale dynamics of the epidemic. Despite these considerations, we acknowledge that several limitations should be considered when interpreting our results. This was a single-center study, which may have restricted the generalizability of the findings compared that with broader or more diverse populations. Moreover, while we normalized the data by month to account for variability in sample numbers over time, this approach may have still introduced bias when considering months with a low sampling density. Specifically, mandatory medical surveillance in the AOUI Hospital of Verona, which involved the screening of medical personnel for SARS-CoV-2, ended at the end of March 2023—halfway through the study. The reduction in the number of samples processed following the change in surveillance protocol could have impacted the statistical depth of the study. Lastly, the lack of patient demographic data do not permit the correlation between viral variants and host factors such as age, sex, or comorbidities. To validate and expand upon our observations, more multi-center studies are necessary, including diverse cohorts and comprehensive metadata.
Despite these limitations, our findings contribute valuable insights to the ongoing monitoring of SARS-CoV-2 evolution. Continuous genomic surveillance, within a collaborative international framework, remains essential for the early detection of variants of concern and for the timely adaptation of vaccines and therapeutics [117,118]. In summary, this study underscores the importance of sustained, integrative surveillance to anticipate and mitigate the impact of emerging SARS-CoV-2 variants.”
Comment 2: The added comment about global trends being mirrored is somewhat vague (lines 601-602). It would be helpful to have at least some exemplary discussion or clearer point (e.g., is there a lag? Are there continental differences? Are diversity patterns generally simiar?).
Response 2: The comment was contextualized with figures (new Figure 4 and Supplementary Figure S2) and detailed accordingly:
Lines 139-145: “The mutation trends described were manually verified using CoV-Spectrum. Additionally, Spearman’s correlation analyses were performed to compare the global trend of each mutation with that observed in this study. The results are shown in Figure 4 (panels A, B, C, and D). Almost all substitutions show a positive correlation index. Mutations with values below 0.1 were further investigated and despite their low correlation, similar tendencies between local and global monthly prevalences can be observed. Supplementary Figure S2 compares their local and global trends.”
Comment 3: The authors should have at least some broader conclusions from their data that would help pull bring together the diverse results that are being presented and add analysis to what is currently still a somewhat descriptive overview of and help the user synthesize key observations. For instance, the authors could consider adding a summary figure or table that highlights a subset of particularly impactful mutations (e.g., those affecting immune escape, receptor binding, or showing convergent evolution). This would help orient readers amid a large volume of data.
Response 3: We added an additional figure (Figure 14) and a supplementary table (Table S1). We also added the following section accordingly:
Lines 557-572: “Functional analysis of the 88 identified mutations reveals that immune evasion represents the most common adaptive trait, with 56.8% (49 out of 88) of the mutations associated with increased escape from host immunity. This pattern is particularly noticeable in emerging and transient categories, which together account for 70% (35 out of 50) of immune-evasive mutations. The divergent group, although smaller, shows a similar trend, with six out of eight mutations enhancing immune evasion. Only one divergent mutation (G446S) is associated with reduced immune evasion. These findings suggest that immune escape undergoes a dominant selective pressure.
By comparison, mutations affecting ACE2 binding affinity represent the second-most frequent involvement, with 12 mutations linked to increased binding—primarily within the emerging group with 6 mutations—and 9 associated with reduced binding, mostly among transient mutations (5 mutations). Interestingly, six of the mutations that reduce ACE2 affinity co-occur with increased immune evasion and one with increased transmissibility, as represented in Figure 14. This may imply that immune escape could be prioritized over receptor binding optimization in certain evolutionary contexts. The list of these mutations is reported in Supplementary Table S1.”
Comment 4: There are still some language issues remaining in the paper. For example, in the revision in lines 600-603 describes Verona as a “relatively small reality.” Please make sure to review the material added in the revision to make sure that has been adequately edited.
Response 4: The overall fluidity of all the edited parts was improved. Line 640 specifically was changed to “relatively small setting” (Response 1). We would also like to kindly note that the English throughout the entire manuscript has been improved with the support of the journal's proofreading service.
